# Doubly linked chiral phenanthrene oligomers for homogeneously π-extended helicenes with large effective conjugation length

Yusuke Nakakuki[1], Takashi Hirose [2,3]✉, Hikaru Sotome [4], Min Gao [5], Daiki Shimizu [1], Ruiji Li [1,6], Jun-ya Hasegawa [5], Hiroshi Miyasaka [4] & Kenji Matsuda [1]✉

Helically twisted conductive nanocarbon materials are applicable to optoelectronic and electromagnetic molecular devices working on the nanometer scale. Herein, we report the synthesis of per-*peri*-perbenzo[5]- and [9]helicenes in addition to previously reported π-extended [7]helicene. The homogeneously π-extended helicenes can be regarded as helically fused oligo-phenanthrenes. The HOMO−LUMO gap decreased significantly from 2.14 to 1.15 eV with increasing helical length, suggesting the large effective conjugation length (ECL) of the π-extended helical framework. The large ECL of π-extended helicenes is attributed to the large orbital interactions between the phenanthrene subunits at the 9- and 10-positions, which form a polyene-like electronic structure. Based on the experimental results and DFT calculations, the ultrafast decay dynamics on the sub-picosecond timescale were attributed to the low-lying conical intersection.

[1] Department of Synthetic Chemistry and Biological Chemistry, Graduate School of Engineering, Kyoto University, Katsura, Nishikyo-ku, Kyoto 615-8510, Japan. [2] Institute for Chemical Research, Kyoto University, Uji, Kyoto 611-0011, Japan. [3] PRESTO, Japan Science and Technology Agency (JST), 4-1-8 Honcho, Kawaguchi, Saitama 332-0012, Japan. [4] Division of Frontier Materials Science and Center for Promotion of Advanced Interdisciplinary Research, Graduate School of Engineering Science, Osaka University, Toyonaka, Osaka 560-8531, Japan. [5] Institute for Catalysis, Hokkaido University, Sapporo, Hokkaido 001-0021, Japan. [6] School of Pharmacy, Jining Medical University, 669 Xueyuan Road, Rizhao, Shandong 276800, China. ✉email: hirose@scl.kyoto-u.ac.jp; kmatsuda@sbchem.kyoto-u.ac.jp

Helical structures are chiral structural motifs that are found universally in nature, and scientists have been curious about their functions derived from the unique asymmetric structures[1]. Helically twisted polycyclic aromatic hydrocarbons (HPAHs), represented by helicenes[2], are one of the major targets in synthetic chemistry with potential applications in asymmetric catalysis[3], optoelectronics[4], and nonlinear optical materials[5]. The unique properties derived from helical molecular structures, such as selective response to circularly polarized lights[6,7], chirality-induced spin selectivity[8,9], and mechanical structural changes that behave as molecular springs[10], have been extensively investigated from both experimental and theoretical perspectives. The availability of HPAHs with high electric conductivity will open up further potential applications; for example, helical graphenes are theoretically considered as promising candidates for efficient nanometer-sized molecular solenoids[11].

Since carbo[6]helicene ($C_{26}H_{16}$) was synthesized by Newman and Lednicer in 1956[12], many synthetic chemists have been engaged in the synthesis of various HPAHs. The longest carbo-helicene, i.e., carbo[16]helicene ($C_{66}H_{36}$), was synthesized by Murase and Fujita in 2015[13]. π-Expanded helicenes with large helical diameters have been reported by Vollhardt ($C_{42}H_{18}$) in 2002[14,15], Tilley ($C_{96}H_{96}$) in 2017[16], and our group ($C_{54}H_{30}$) in 2018[10]. The recent success in the synthesis of π-extended HPAHs with large helical widths represents a remarkable development in organic chemistry. Morin reported in 2017 a pioneering work on doubly linked triphenylene polymers, which are regarded as helically coiled graphene nanoribbons[17]. Helical bilayer nanographene ($C_{98}H_{30}(^tBu)_{10}$) was successfully synthesized by Martín in 2018[18]. Campaña reported in the same year a hexa-peri-hexabenzocoronene-based π-extended [7]helicene ($C_{114}H_{30}(CO)_2(^tBu)_8$) that is a undecabenzo[7]superhelicenes derivative with two carbonyl group embedded at the ends[19]. Wang reported the synthesis of undeca-benzo[7]superhelicenes ($C_{114}H_{26}(^tBu)_{12}$) without carbonyl group in 2021[20]. Although various π-extended HPAHs have been synthesized, the HPAHs reported so far share a common feature that the absorption edge ($\lambda_{edge}$) does not exceed 650 nm (1.91 eV) despite their relatively large molecular structures, suggesting that increasing the effective conjugation length (ECL) of HPAHs with distorted molecular structures is a challenging issue.

Considering the transport of charge carriers along a covalently bonded organic π-conjugated system, large interactions between the π-electrons of adjacent units are the key to good carrier conductivity. In this context, π-conjugated molecules with large ECLs are potentially applicable to conductive materials in the nanometer scale, which are called molecular wires[21,22]. Several promising molecular structures have been developed as molecular wires, including doubly linked naphthalene oligomers, or rylenes, by Müllen and Wu ($\lambda_{edge}$ ~2500 nm)[23,24], triply linked porphyrin tapes by Osuka ($\lambda_{edge}$ > 2500 nm)[25,26], polyynes by Lambert and Tykwinski ($\lambda_{edge}$ ~700 nm)[27–29], and carbon-bridged oligo(p-phenylenevinylene)s by Tsuji and Nakamura ($\lambda_{edge}$ ~550 nm)[30,31]. The large ECLs observed in the molecular wires are likely attributed to a strong electronic coupling between adjacent repeating units; the energy gap between the HOMO and LUMO is greatly reduced when there are large in-phase (bonding) and out-of-phase (anti-bonding) interactions in each of the LUMOs and HOMOs, respectively (Fig. 1a). Thus, the concepts and examples of molecular wires with linear and planar structures have been developed; however, chiral molecular wires with helical structures are hardly developed at the moment.

In this work, we report that a class of doubly linked helical phenanthrene oligomers has a significant increase in ECL with increasing molecular length, which is a promising candidate for the helically twisted chiral molecular wires (Fig. 1b). Given that phenanthrene has the largest orbital coefficients at the 9- and 10-positions (i.e., the convex armchair edge with a significant C=C double bond character, the so-called K-region[32]), we conceived that the strongest electronic coupling can be obtained when the K-region of phenanthrene are covalently connected together, which gives the helical phenanthrene oligomers. The 1-to-1′ 10-to-10′ doubly linked helical phenanthrene oligomers thus designed have a significantly smaller HOMO–LUMO gap than the linear structural isomer in which the K-region is differently connected to the 1 or 8 positions of phenanthrene (see section 2.1 of Supplementary Information for details). The systematic synthesis of the doubly linked phenanthrene-based π-extended HPAHs with the different molecular length—i.e., tetra-peri-tetrabenzo[5]helicene (**1**, $C_{34}H_{16}(CH_3)_2$), hexa-peri-hexabenzo[7]helicene (**2**, $C_{48}H_{24}$)[33], and octa-peri-octabenzo[9]helicene (**3**, $C_{62}H_{30}$)—enabled us to investigate the electronic and photophysical properties of the helically twisted nanographene molecules with remarkably large ECLs. It is noted that **3**, with only 62 π-electrons, has a low-energy absorption edge that reaches the near-infrared region ($\lambda_{edge}$ = 1020 nm).

## Results and discussion

**Synthesis of the π-extended HPAHs.** The synthesis of the π-extended [7]helicene **2** has been previously reported by our group[33]. The key synthetic strategy of the homogeneously π-extended HPAH was first to form a partially saturated ring-substituted [7]helicene—i.e., hexa-peri-hexa(1,3-propylene)[7]helicene (**5**)—by a photochemical reaction followed by a dehydrogenative aromatization reaction (Fig. 2a), which is an efficient approach to create π-conjugated molecules with highly strained helical geometry. In applying the same strategy to the π-extended [9]helicene **3**, 2,3,4-trihydro-5,10-(1,3-propylene)anthracene-1-one (**7**) was designed as a π-extended terminal subunit (Fig. 2b). Compound **7** was synthesized from 1-formyl-2-methoxynaphtharene over nine reaction steps in ca. 25% total yield (see section 1 of Supplementary Information for details). A 2-fold McMurry coupling reaction of dodecahydrodibenzo[b,n]perylene-1,14-dione (**6**)[33] with excess amount of **7** afforded a 3,6-bis(β-(naphthalene-2-yl)vinyl)phenanthrene derivative **8** in 53% yield. Subsequently, photocyclodehydrogenation of **8** in the presence of iodine and propylene oxide followed by dehydrogenative aromatization using 2,3-dichloro-5,6-dicyano-1,4-benzoquinone (DDQ) provided the target compound **3** in 6% yield in the two steps.

[5]Helicene is known to undergo an annulation reaction upon photoirradiation, which produces a planar PAH, i.e., benzo[ghi]perylene[34]. To prevent undesired overannulation during photochemical reaction steps, we intentionally introduced methyl groups at the inner position of the π-extended [5]helicene **1** (Fig. 2c)[35]. To this end, compound **10**, that is the 8-methyl derivative of **7**, was prepared from 1-formyl-4-methylnaphthalene through nine steps in ca. 8% total yield (see section 1 of Supplementary Information for details). Dimerization of **10** by a McMurry coupling reaction afforded **11** in 95% yield, and subsequent photocyclodehydrogenation of **11** gave 1,14-dimethyl-tetra-peri-tetra(1,3-propylene)[5]helicene (**12**) in 41% yield. No overannulation was detected during the photochemical reaction of **11**. The dehydrogenative aromatization of **12** successfully proceeded, affording a mixture of **1** together with its unexpected closed-ring isomer **1c** in 25% yield. Interestingly, **1** and **1c** showed a reversible photoisomerization behavior; **1** isomerized to **1c** upon irradiation with violet light at 400 nm, and **1c** isomerized back to **1** upon irradiation with UV light at 337 nm (see section 2.2 of Supplementary Information for details). The two isomers **1** and **1c** were separated by high-performance liquid chromatography (HPLC) under dark conditions, and no thermal isomerization between **1** and **1c** was observed at room temperature.

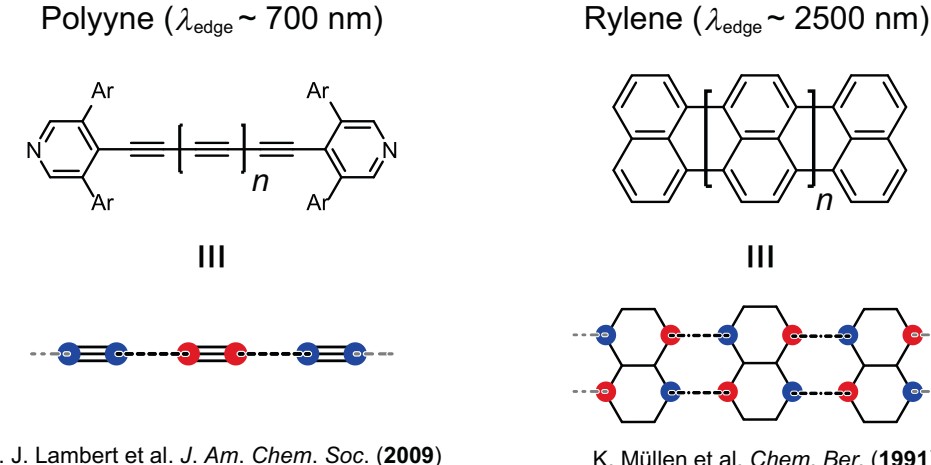

**a** Linear Molecular Wires

Polyyne ($\lambda_{edge} \sim$ 700 nm)

C. J. Lambert et al. *J. Am. Chem. Soc.* (**2009**)
R. R. Tykwinski et al. *Nat. Chem.* (**2020**)

Rylene ($\lambda_{edge} \sim$ 2500 nm)

K. Müllen et al. *Chem. Ber.* (**1991**)
W. Jishan et al. *Chem.* (**2017**)

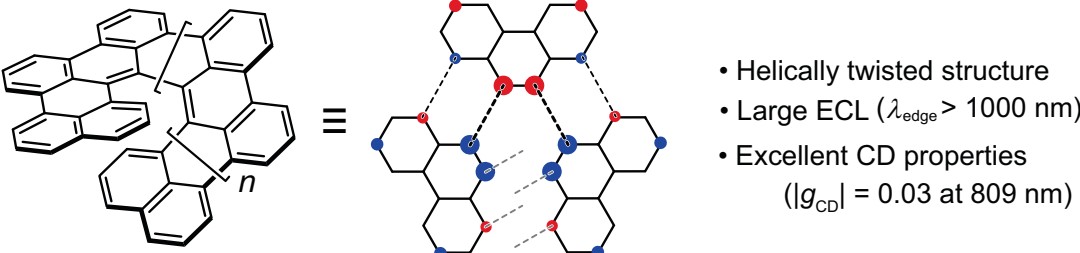

**b** Helical Molecular Wire (This Work)

- Helically twisted structure
- Large ECL ($\lambda_{edge}$ > 1000 nm)
- Excellent CD properties
  ($|g_{CD}|$ = 0.03 at 809 nm)

**Fig. 1 Molecular structures and orbital interactions of molecular wires. a** Linear molecular wires. **b** Helical molecular wires. The out-of-phase (anti-bonding) orbital interactions generate the HOMO of the resulting molecular wires. Blue and red circles indicate the orbital coefficients in the opposite phase, seen in the HOMO of the repeating unit.

The structures of **1** and **3** were fully characterized by [1]H NMR, [13]C NMR, HMQC/HMBC measurements, and high-resolution mass spectrometry (see section 3 of Supplementary Information for details). π-Extended helicenes **1**–**3** were stable under ambient conditions in solution and in the solid state for more than one month.

**Bond length and aromaticities of the π-extended HPAHs**. Our group has previously reported the single crystal structure of **2**[33]. In this work, single crystals of **1** and **3** suitable for X-ray analysis were obtained by a slow vapor diffusion of methanol into a concentrated solution of **1** in chloroform and a slow evaporation of toluene solution of **3**, respectively (Fig. 2). The C–C bond lengths along the inner helical rims were 1.47–1.48 Å for the even-numbered rings (i.e., rings B, D, and so on), which are significantly longer than the values for the odd-numbered rings (i.e., 1.39–1.40 Å for rings A, C, E, and so on) (Fig. 3a). The large bond length alternation was commonly found in the inner helical rims of **1**–**3**, suggesting a polyene-like electronic structure inside their helical skeleton. According to the nucleus independent chemical shifts (NICS)[36] calculations of **1**–**3**, the even-numbered rings have small NICS(1) values (i.e., −0.31 to 1.93 ppm for the rings B, D, and so on), suggesting the non-aromatic characters of the six-membered rings that doubly link the phenanthrene subunits (Fig. 3b). Furthermore, moderate NICS(1) values were

found on the middle rings of the phenanthrene subunits (i.e., −6.24 to −4.61 ppm for the rings C and so on), while negatively large NICS(1) values on the terminal and outer rings (i.e., −9.83 to −7.22 ppm for the rings A, K, L, and so on). The NICS values of **1**–**3** suggested a characteristic electronic structure that can be explained as a phenanthrene oligomers terminally capped with the naphthalene-1,8-diyl subunits, which is consistent with the bond lengths determined by the X-ray crystallography and the anisotropy of current-induced density[37] calculations (see section 2.4 of Supplementary Information for details). The observed electronic structure is also reasonably explained by the Clar's sextet theory[32]; the number of the Clar's sextet circles is maximized when they are located at the outer rings. As a result, the sp[2] carbon atoms at the 9- and 10-positions of phenanthrene subunits (i.e., the K-region) form a polyene-like structure along the inner helical rims with all-($Z$) configuration (Fig. 3c).

**UV-vis-NIR absorption and CD spectra**. Figure 4a shows the UV-vis-NIR absorption spectra of π-extended helicenes **1**–**3** in toluene. Compounds **1**–**3** showed the lowest-energy absorption bands at 500–1020 nm ($\varepsilon_{max}$ = 19000 $M^{-1}$ $cm^{-1}$ at 529 nm for **1**, 4700 $M^{-1}$ $cm^{-1}$ at 675 nm for **2**[33], and 2500 $M^{-1}$ $cm^{-1}$ at 814 nm for **3**). With increasing helical length, the absorption maxima of the π-extended HPAHs were prominently red-shifted, which is in marked contrast to the unsubstituted carbo[$n$]

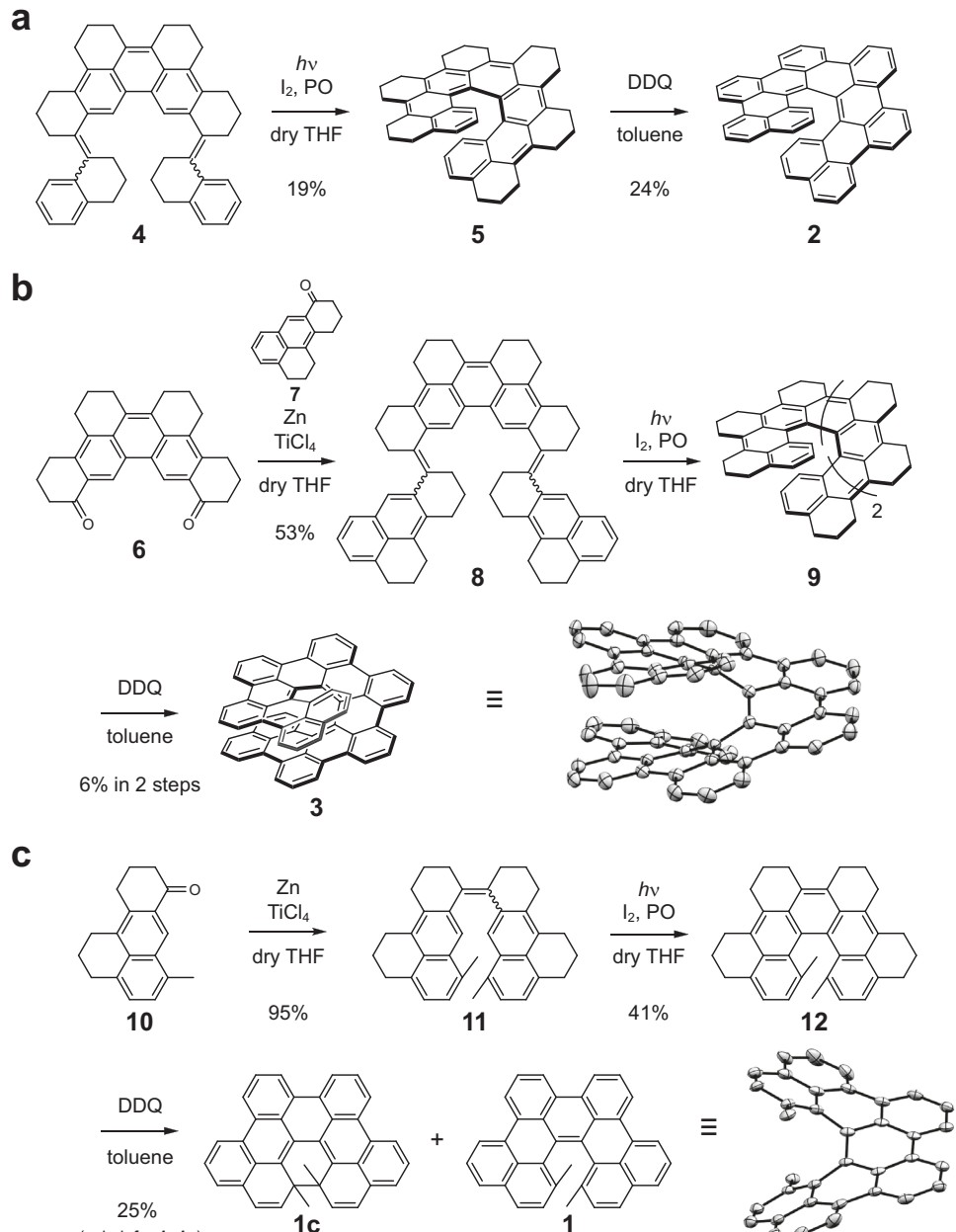

**Fig. 2 Synthesis of π-extended helicenes. a** π-extended [7]helicene **2**[33]. **b** [9]helicene **1**. **c** [5]helicene **3**. The ORTEP drawings of **1** and **3** recorded at 143 and 100 K, respectively, showing 50% probability thermal ellipsoids. Hydrogen atoms are omitted for clarity in the ORTEP drawings.

helicenes; the long-wavelength absorption edges of carbo[n]helicenes appear below 500 nm, even for the longest derivative, i.e., [16]helicene (C$_{66}$H$_{36}$)[7,13]. To the best of our knowledge, the absorption edge of **3** ($\lambda_{edge}$ = 1020 nm) is the longest wavelength among the helical HPAHs in the neutral closed-shell electronic state reported so far[24,32], suggesting that the π-extended HPAHs **1–3** is a characteristic chemical structure that shows the large ECL while having a helically distorted chiral geometry.

According to the time-dependent density functional theory (TD-DFT) calculations at the B3LYP/6-311G(2d,p) level of theory, the characteristic lowest-energy absorption bands were attributed to the HOMO → LUMO transitions ($\lambda_{ex,calc}$ = 552, 752, and 899 nm for **1**, **2**, and **3**, respectively). The HOMO and LUMO of **1–3** are delocalized throughout the helical molecular framework, and relatively large orbital coefficients were found at the inner helical rim with increasing helical length (Fig. 4c). The

elongation of ECLs was quantitatively investigated by plotting the lowest singlet excitation energies (i.e., for the S$_0$ → S$_1$ transition) of carbo[n]helicenes, rylenes, and the π-extended HPAHs against the reciprocal of the numbers of π-electrons (1/$N_{el}$) (Fig. 4b). The excitation energies of carbo[n]helicenes approached about 2.5 eV with increasing helical length, suggesting no significant elongation of ECL in the conventional helicenes. Interestingly, the excitation energies of the π-extended HPAHs approached about 1.0 eV, and the decrease in excitation energies was comparable to that of rylenes, indicating that the doubly linked phenanthrene oligomers are expected to exhibit properties as the helical molecular wires.

Electrochemical measurements revealed the redox activities of the π-extended HPAHs **1–3**, reflecting their small HOMO–LUMO gap. The π-extended [5]helicene **1** showed an oxidation potential at 0.34 V and reduction potential at −2.01 V vs. Fc/Fc$^+$ in dry

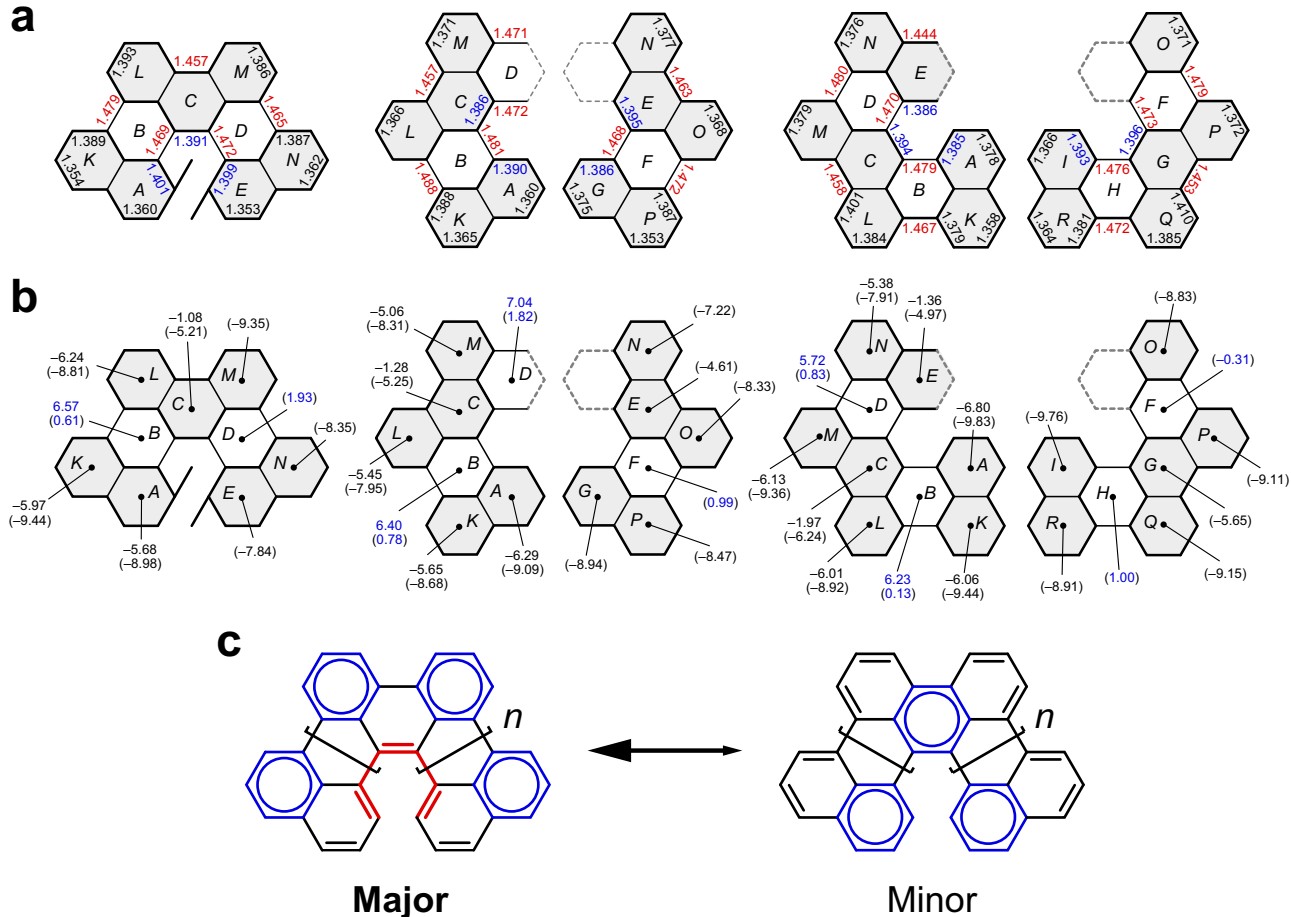

**Fig. 3 Bond length and local aromaticity. a** Representative bond lengths of **1** (left), **2** (middle)[33], and **3** (right) in angstrom determined by X-ray crystallography. The ring structures highlighted in gray represent phenanthrene and naphthalene substructures. **b** NICS(0) values of (P)-**1**, **2**, and **3** calculated at the GIAO-B3LYP/6-311G(2d,p) level of theory. NICS(1) values are shown in parentheses. The NICS(1) value was calculated at a vertical distance of 1 Å from the centroid of each benzene ring, in the front direction from the paper. **c** Resonance structures of the π-extended HPAHs based on the doubly linked phenanthrene oligomers. Benzenoid rings are highlighted in blue and the polyene-like electronic structure found in the inner helical rim is highlighted in red.

THF, which leads to an electrochemical HOMO–LUMO gap ($\Delta E_{el}$) of 2.35 eV. The π-extended [7] and [9]helicenes **2** and **3** showed oxidation potentials at 0.09 and −0.10 V and reduction potentials at −1.71 and −1.63 eV, respectively, and the $\Delta E_{el}$ values were estimated as 1.80 and 1.53 eV for **2** and **3** (Table 1). According to the DFT calculations, their HOMO–LUMO gaps ($\Delta E_{DFT}$) were determined to be 2.55, 2.04, and 1.83 eV for **1**, **2**, and **3**, respectively (Table 1), which is in good agreement with experimentally determined $\Delta E_{el}$ as well as $\Delta E_{opt}$ determined by the absorption edges ($\lambda_{edge}$).

Enantiomers of π-extended helicenes were successfully separated by chiral HPLC (see section 2.7 of Supplementary Information for details) and their circular dichroism (CD) spectra were recorded in toluene (Fig. 4a). Based on the comparison of the sign of CD spectra with the result of TD-DFT calculations, the chiralities of π-extended helicenes were determined as follows: the (P)- and (M)-isomers of **1**–**3** commonly show the first positive and negative CD signals, respectively. The dissymmetry factor of absorption (i.e., $g_{CD}$ defined as $g_{CD} = \Delta\varepsilon/\varepsilon$) was found to improve significantly with increasing helical length; the $|g_{CD}|$ values were 0.0020 (at 529 nm), 0.016 (at 680 nm), and 0.030 (at 809 nm) for **1**, **2**, and **3**, respectively, and the $g_{CD}$ value of **3** was more than 10 times that of **1**. It is noted that the CD signal observed in the near-infrared region (i.e., 780–1020 nm) is rare, and the large $g_{CD}$

value on the order of $10^{-2}$ is relatively high compared to the recently reported values for organic molecules[38,39].

The thermal stability of the isolated enantiomers is an important aspect when considering the applications as chiral molecular wires. No racemization of **2** and **3** between the (P)- and (M)-isomers was detected in toluene at 90 °C for 7 h (see section 2.8 of Supplementary Information for details). Unfortunately, the racemization process of **1** was hard to investigate due to photoisomerization caused by the probe light during CD measurements. Based on DFT calculations at the B3LYP/6-311G(2d,p) level, the activation barriers of helical inversion ($\Delta G^{\ddagger}$) were calculated to be 169.3, 187.3, and 180.8 kJ mol$^{-1}$ for **1**, **2**, and **3**, respectively, indicating their long half-lives of enantiomers longer than one hundred years at 298 K. The high configurational stability of π-extended helicenes is comparable to that of carbo[n]helicenes: i.e., $\Delta G^{\ddagger} = 184.9$, 174.5, and 182.0 kJ mol$^{-1}$ experimentally determined for 1,14-dimethylated [5] helicene derivative, [7]helicene, and [9]helicene, respectively[40,41].

**Ultrafast dynamics in the excited state.** In the previous work, we reported that the π-extended [7]helicene **2** showed an ultrafast de-excitation dynamics from the $S_1$ to $S_0$ states, with the lifetime of only $\tau_{S1} = 1.2$ ps, which is 4 orders of magnitude shorter than

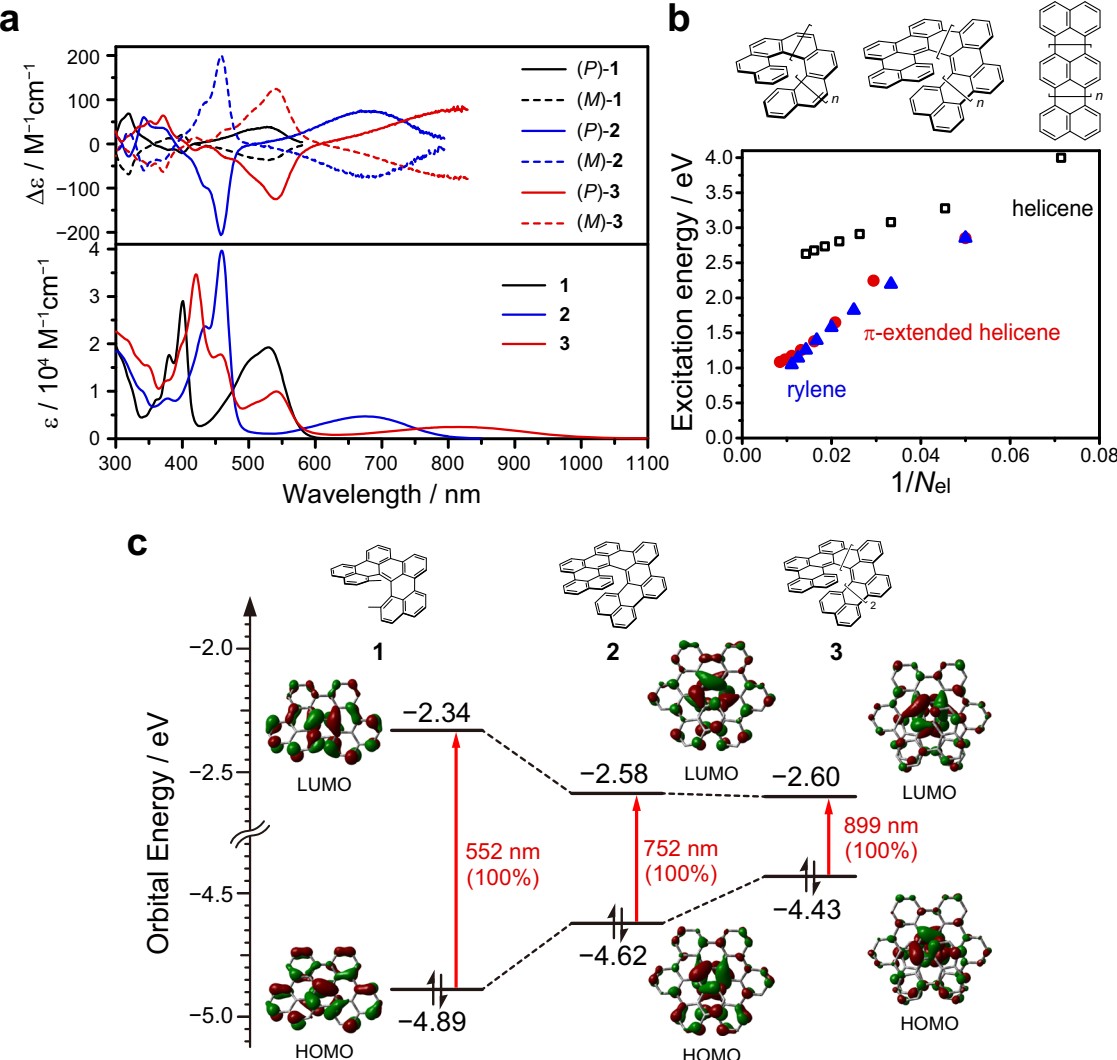

**Fig. 4 Photoabsorption properties. a** UV–vis–NIR absorption and circular dichroism (CD) spectra of **1** (black), **2** (blue), and **3** (red) in toluene at 25 °C. In the CD spectra, the solid and dashed lines represent the (*P*)- and (*M*)-isomers, respectively. **b** Excitation energies for the lowest energy ($S_0 \rightarrow S_1$) transition of helicenes (black square), π-extended helicenes (red circle), and rylenes (blue triangle) calculated at the TD-RB3LYP/6-311G(2d,p) level of theory. **c** Orbital correlation diagram of **1–3** calculated at the TD-RB3LYP/6-311G(2d,p) level of theory.

**Table 1 Summary of the optoelectronic properties of 1, 2, and 3.**

| Compd. | optical[a] | | electrochemical[b] | | | theoretical[c] |
|---|---|---|---|---|---|---|
| | $\lambda_{edge}$/nm | $\Delta E_{opt}$/eV | $E_1^{ox}$/V | $E_1^{red}$/V | $\Delta E_{el}$/eV | $\Delta E_{DFT}$/eV |
| **1** | 580 | 2.14 | 0.34 | −2.01 | 2.35 | 2.55 |
| **2** | 800 | 1.55 | 0.09 | −1.71 | 1.80 | 2.04 |
| **3** | 1020 | 1.22 | −0.10 | −1.63 | 1.53 | 1.83 |

$\lambda_{edge}$ absorption edge determined at 10% of maximum absorption; $\Delta E_{opt}$ optical energy gap determined by $\lambda_{edge}$; $E_1^{ox}$ the first oxidation potential; $E_1^{red}$ the first reduction potential; $\Delta E_{el}$ electrochemical energy gap determined by $E_1^{ox}$ and $E_1^{red}$; $\Delta E_{DFT}$ the HOMO–LUMO gap determined by DFT calculations.
[a]Based on absorption spectra measured in toluene.
[b]Measured in THF with *n*-Bu₄NPF₆ (0.1 M) on a platinum electrode (vs Fc/Fc⁺). Scan rates were 0.05 V s⁻¹.
[c]Calculated at the RB3LYP/6-311G(2d,p) level of theory.

that of unsubstituted [7]helicene ($\tau_{S1} = 14$ ns), however, the photophysical cause of this ultrafast dynamics was not clear at the time[33]. In this work, we investigated the excited-state dynamics of the π-extended [5] and [9]helicene derivatives **1** and **3**, and found

that the remarkably short $S_1$-state lifetime on the order of picoseconds is a characteristic property of the π-extended HPAHs **1–3**, which is observed regardless of the helical length (Fig. 5a).

Based on the analyses of time profiles (see section 2.9 of Supplementary Information for details), the $S_1$-state lifetimes for **1** and **3** were determined as $\tau_{S1} = 0.76$ and 0.38 ps, respectively, and the time constants of the vibrational cooling in the ground state were $\tau_{vc} = 8.5$ and 4.9 ps for **1** and **3**, respectively. The determined time constants for **1** and **3** are comparable to those reported for **2** (i.e., $\tau_{S1} = 1.2$ ps and $\tau_{vc} = 9.7$ ps)[33]. The nonradiative decay rate for the $S_1 \rightarrow S_0$ transition ($k_{nr}$) was determined as $k_{nr} = 1.3 \times 10^{12}$, $8.3 \times 10^{11}$, and $2.6 \times 10^{12}$ s⁻¹ for **1**, **2**, and **3**, respectively, which are 4 orders of magnitude faster than those of unsubstituted carbohelicenes, i.e., $k_{nr} = 3.7 \times 10^7$, $7.0 \times 10^7$, and $1.0 \times 10^8$ s⁻¹ for [5], [7], and [9]helicenes, respectively[42]. It is noted that recently reported π-extended helicenes, which do not have the polyene-like electronic structure along the inner helical rim, exhibit emission properties with the $k_{nr}$ values on the order of $10^7$ to $10^8$ s⁻¹ (see Table S8 in Supplementary Information for details).

The ultrafast excited state dynamics of **1–3** suggests that the electronic excited states are likely de-activated via a conical

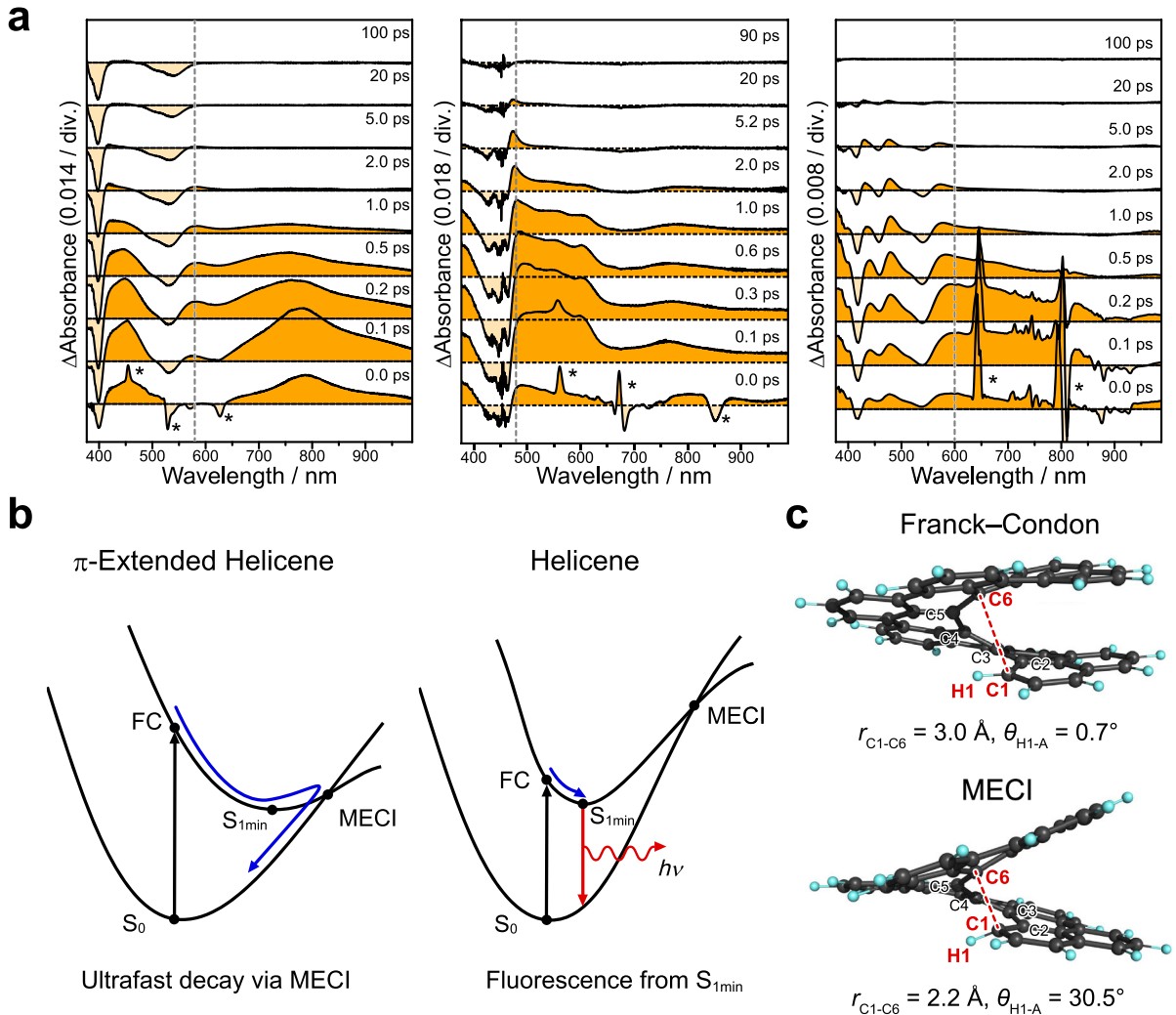

**Fig. 5 Ultrafast dynamics in the excited state. a** Transient absorption (TA) spectra of **1** (left), **2** (middle)[33], and **3** (right) recorded in toluene. Positive and negative bands are highlighted in orange and pale orange, respectively. Signals denoted by asterisks are due to stimulated Raman scattering of solvent. **b** Schematic illustration of excited-state dynamics of π-extended helicenes and helicenes. The black, blue, and red arrows represent excitation, nonradiative decay, and radiative decay processes, respectively. **c** Franck–Condon (FC) and minimum energy conical intersection (MECI) geometries of π-extended [7] helicene **2** optimized using SF-ωB97XD/6-31G(d) level of theory.

intersection (CI)[43,44] that is located near the vertically excited Franck−Condon (FC) state. To investigate the photophysical cause of the ultrafast dynamics of the π-extended HPAHs, the minimum energy CI (MECI) structures in the $S_1$ state were calculated using the spin-flip TD-DFT method[45,46]. The relative energy of the MECI against FC ($\Delta E_{\text{MECI−FC}}$) were found to be negative for the π-extended HPAHs (i.e., $\Delta E_{\text{MECI−FC}} = -0.63$, $-0.17$, and $-0.06$ eV for **1**, **2**, and **3**, respectively), which is in marked contrast to the positive $\Delta E_{\text{MECI−FC}}$ values for the conventional carbo[$n$]helicenes (i.e., $\Delta E_{\text{MECI−FC}} = +0.63$, $+0.50$, and $+0.85$ eV for [5], [7], and [9]helicene, respectively). The negative $\Delta E_{\text{MECI−FC}}$ values of **1**–**3** suggest that the energy of the MECI structures are lower than those of FC, where the excited state of the π-extended HPAHs can be de-activated via MECI immediately after photoexcitation (Fig. 5b). Figure 5c shows the structural change from the FC to MECI states of the π-extended [7]helicene **2**. The structural relaxation in the $S_1$ state is characterized by (i) the significant decrease in the C···C distance between C1 and C6 ($r_{\text{C1···C6}}$) from 3.0 to 2.2 Å, and (ii) the marked increase in the tilt angle of H1 relative to the least squares plane of the terminal ring A ($\theta_{\text{H1−A}}$) from 0.7 to 30.5° (Fig. 5c). It

is noted that the nonradiative deactivation dynamics of **2** and **3** are comparable to that of **1** showing photochemical cyclization to **1c**[47,48], and the structural change from the FC to MECI structures is reminiscent of the electrocyclic reaction of 1,3,5-hexatriene[49]. Thus, the ultrafast decay dynamics of π-extended HPAHs **1**–**3** can be explained by their low-lying MECIs compared to the FC states that are likely derived from the polyene-like electronic structure found in the inner helical rim of the doubly linked chiral phenanthrene oligomers.

In summary, we have synthesized π-extended [5]- and [9]helicenes in addition to previously reported π-extended [7]helicene, and have proved that the combination of photo-cyclization followed by aromatization reactions is an effective strategy for the synthesis of the helically twisted π-extended HPAHs. The outstandingly large ECL observed for the π-extended helicenes indicates that phenanthrene (the number of benzene rings, $n_{\text{ring}} = 3$), which has an angularly annulated PAH framework, is an essential repeating unit in the creation of a helical molecular wire, in contrast to naphthalene ($n_{\text{ring}} = 2$) that forms a linear molecular wire. The large ECL and ultrafast excited state dynamics of the π-extended HPAHs **1**–**3** can be attributed to

the significant contribution of the polyene-like structure consisting of the K-region of the phenanthrene repeating units. The established molecular design of the chiral π-conjugated framework leads to the creation of helically twisted molecular wires. The development of helical molecular wires with sufficient length is of potential interest, as graphene nanoribbons with sufficient length have been experimentally achieved by on-surface synthesis. The creation of other π-extended HPAHs with different spatial distributions of frontier orbitals, e.g., HPAHs having a large ECL with their frontier orbitals localized at the outer rim moieties, will expand the scope of the chemistry of helical molecular wires. Further researches on magnetic and spintronic devices based on the helically twisted nanocarbon materials are currently ongoing in our group.

## Methods

**Synthesis of 1,14-bis(1,2,3,9,10,11-hexahydrobenzo[d,e]anthracen-8-ylidene)-2,3,4,5,6,7,8,9,10,11,12,13-dodecahydrodibenzo[b,n]perylene (8)**. To a suspension of zinc powder ($4.5 \times 10^2$ mg, 6.9 mmol) in dry THF (8.5 mL) was added $TiCl_4$ ($7.9 \times 10^2$ mg, 4.2 mmol). A solution of dodecahydrodibenzo[b,n]perylene-1,14-dione (6, 78 mg, 0.20 mmol) and 2,3,4-trihydro-5,10-(1,3-propylene)anthracene-1-one (7, $4.5 \times 10^2$ mg, 1.9 mmol, 9.6 eq.) in dry THF (25 mL) were then added to the reaction mixture. The resulting solution was refluxed for 2 h and then was cooled to room temperature. The reaction was quenched by adding aq. HCl (1 M, ca. 15 mL), and the resulting solution was extracted with $CH_2Cl_2$ (ca. 150 mL) and washed with water. The organic layer was dried over $MgSO_4$, filtrated, and concentrated in vacuo. The residue was purified by silica gel column chromatography (hexane/$CH_2Cl_2$ = 85/15 to 70/30) to remove the dimer of 7, affording compound 8 (84 mg, 0.10 mmol, 53%) as a yellow solid. [1]NMR spectrum of the product was very complicated probably due to the mixture of *trans/cis* isomers, which hampered further assignment (Supplementary Figure 53). Note that the production of 8 was confirmed by LR-MALDI-TOF mass spectrometry. LRMS–MALDI–TOF (m/z): [M]$^+$ calcd for $C_{62}H_{58}^+$, 802; found, 802.

**Synthesis of octa-peri-octabenzo[9]helicene (3)**. A round-bottom quartz flask was charged with 8 (68 mg, 0.085 mmol), iodine (107 mg, 0.42 mmol), and dry THF (90 mL). The solution was purged with $N_2$ bubbling for 5 min. After the addition of propylene oxide (10 mL, ca. 150 mmol), the solution was further purged with $N_2$ bubbling for 2 min. The solution was stirred and irradiated with a super-high-pressure mercury lamp (500 W) through a sharp-cut filter UV-29 for 75 min. The mixture was stirred with $Na_2S_2O_3$ powder, filtered, and then was concentrated in vacuo. The crude product was passed through a plug of silica gel (hexane/dichloromethane = 80/20) to give crude 9 (13 mg) as a yellow solid. A solution of crude 9 (13 mg) in toluene (13 mL) was degassed by $N_2$ bubbling at 90 °C for 5 m. After the addition of DDQ (44 mg, 0.19 mmol), the resulting solution was stirred at 90 °C for 5 min. Then the solution was cooled and immediately quenched by the addition of aq. $Na_2S_2O_3$ (conc., ca. 10 mL). The reaction product was extracted with $CH_2Cl_2$ (ca. 100 mL) and washed with water. The combined organic layer was dried over $MgSO_4$ and was passed through a plug of silica gel ($CH_2Cl_2$ as an eluent). The solution was concentrated in vacuo. The crude product was purified by silica gel column chromatography (hexane/$CH_2Cl_2$ = 60/40 to 0/100) and recrystallization ($CH_2Cl_2$/MeOH) to give 3 (3.7 mg, 0.0048 mmol, 6%) as a deep red solid. [1]H NMR (600 MHz, $C_2D_2Cl_4$ with a small excess amount of $N_2H_4{\cdot}H_2O$, δ) 6.36 (t, J = 7.5 Hz, 2H), 6.83 (d, J = 7.8 Hz, 2H), 7.18 (t, J = 7.8 Hz, 2H), 7.23–7.30 (m, 6H), 7.40 (t, J = 7.8 Hz, 2H), 7.60–7.65 (m, 4H), 7.75 (d, J = 7.8 Hz, 2H), 7.82 (d, J = 7.8 Hz, 2H), 8.01 (d, J = 7.2 Hz, 2H), 8.13 (d, J = 7.8 Hz, 2H), 8.42 (d, J = 7.8 Hz, 2H), 8.52 (d, J = 7.2 Hz, 2H). [13]C NMR (151 MHz, $C_2D_2Cl_4$ with a small excess amount of $N_2H_4{\cdot}H_2O$, d): 120.9, 121.1, 121.2, 121.6 (2 peaks overlapped), 121.9, 122.2, 123.4, 125.2, 125.5, 125.6, 126.1 (two peaks overlapped), 126.4, 126.5, 126.9, 127.8, 128.0, 128.2, 128.7, 129.1, 129.5, 129.6, 129.67, 129.73, 130.1, 130.2, 130.6, 130.8, 131.3, 133.5; HRMS–APCI–Orbitrap (m/z): [M + H]$^+$ calcd for $C_{62}H_{31}^+$, 775.2420 found, 775.2425.

**Synthesis of 1,14-dimethyl-tetra-peri-tetra(1,3-propylene)[5]helicene (12)**. To a suspension of zinc powder (1.0 g, 15 mmol) in dry THF (50 mL) was added $TiCl_4$ (1.3 g, 6.9 mmol). A solution of 10 (1.0 g, 4.0 mmol) in dry THF (30 mL) was then added to the reaction mixture. The solution was refluxed for 2 h, cooled to room temperature, and then aq. HCl (1 N, ca. 50 mL) was added. The reaction product was extracted with $CH_2Cl_2$ (ca. 200 mL) and washed with water. The organic layer was dried over $MgSO_4$, filtered, and concentrated in vacuo. The crude product was passed through a plug of silica ($CH_2Cl_2$) to give 11 (0.89 g, 1.9 mmol, 95%) as a yellow solid, which was used for the next reaction without further purification. A round-bottom quartz flask was charged with 11 (100 mg, 0.21 mmol), iodine (330 mg, 1.3 mmol), and dry THF (130 mL). The solution was purged with $N_2$ bubbling for 5 min. After the addition of propylene oxide (15 mL, ca. 220 mmol), the solution was further purged with $N_2$ bubbling for 2 min. The solution was stirred and irradiated with a super-high-pressure mercury lamp (500 W) through a sharp-cut filter (UV-29) for 75 min. The resulting solution was stirred with $Na_2S_2O_3$ powder, filtered, and then was concentrated in vacuo. The crude product was purified by silica gel column chromatography (hexane/dichloromethane = 90/10), which was further purified by gel permeation chromatography (GPC) to give 12 (41 mg, 0.088 mmol, 41%) as a yellow solid. [1]H NMR (600 MHz, CDCl$_3$, δ): 0.76 (s, 6H), 2.02–2.13 (m, 4H), 2.20–2.31 (m, 4H), 3.07–3.31 (m, 16H), 6.82 (d, J = 6.0 Hz, 2H), 7.18 (d, J = 6.0 Hz, 2H); [13]C NMR (151 MHz, CDCl$_3$, d): 22.1, 23.1, 23.4, 27.2, 27.7, 28.6, 31.3, 124.1, 124.4, 126.7, 127.0, 127.8, 128.8, 129.1, 129.2, 131.3, 132.0, 132.5; HRMS–APCI–Orbitrap (m/z): [M + H]$^+$ calcd for $C_{36}H_{35}^+$, 467.2733; found, 467.2727.

**Synthesis of tetra-peri-tetrabenzo[5]helicene (1)**. A solution of 12 (41 mg, 0.088 mmol) in toluene (60 mL) was degassed by $N_2$ bubbling at 80 °C for 5 min. After the addition of 2,3-dichloro-5,6-dicyano-1,4-benzoquinone (DDQ, 150 mg, 0.66 mmol), the resulting solution was stirred at 80 °C for 2 min under nitrogen atmosphere. Then the solution was cooled and immediately quenched by the addition of aq. $Na_2S_2O_3$ (conc. ca. 10 mL). The reaction product was extracted with toluene (ca. 100 mL) and the combined organic layer was concentrated in vacuo. The crude product was purified by silica gel column chromatography (hexane/$CH_2Cl_2$ = 90/10 to 85/15) and GPC to give a mixture of 1 and the closed-ring isomer 1c (10 mg, 0.022 mmol, 25%) as a deep red solid. A pure form of 1 was obtained by HPLC (HPLC column, Mightysil Si 60 250 mm-4.6 mm, 5 μm; eluent, hexane/$CH_2Cl_2$ = 95/5; flow rate, 1.0 mL/min; detection wavelength, 500 nm; $R_f$ = 10.0 and 11.6 min for 1c and 1, respectively). HRMS–APCI–Orbitrap (m/z): [M + H]$^+$ calcd for $C_{36}H_{23}^+$, 455.1794 found, 455.1797.

For 1, [1]H NMR (500 MHz, CDCl$_3$, δ): 1.59 (s, 6H), 6.99 (d, J = 8.5 Hz, 2H), 7.54–7.58 (m, 4H), 7.70 (t, J = 7.7 Hz, 2H), 7.74 (d, J = 8.0 Hz, 2H), 8.28 (d, J = 7.5 Hz, 2H), 8.32 (d, J = 7.5 Hz, 2H), 8.53 (d, J = 7.5 Hz, 2H); [13]C NMR (151 MHz, CDCl$_3$, d): 22.0, 120.8, 121.5, 122.3, 125.7, 126.3, 127.0, 127.2, 129.4, 129.96, 130.04, 130.1, 130.3, 130.5, 130.7, 131.6, 132.9, 133.5.

For 1c, [1]H NMR (500 MHz, CDCl$_3$, δ): 1.75 (s, 6H), 6.43 (d, J = 9.5 Hz, 2H), 6.68 (d, J = 10.0 Hz, 2H), 7.17 (d, J = 6.0 Hz, 2H), 7.45 (dd, J = 8.5, 7.0 Hz, 2H), 7.66 (t, J = 8.0 Hz, 2H), 8.38 (d, J = 8.0 Hz, 2H), 8.39 (d, J = 7.0 Hz, 2H), 8.51 (d, J = 8.0 Hz, 2H); [13]C NMR (126 MHz, CDCl$_3$, d): 26.3, 44.2, 121.4, 122.5, 122.6, 124.7, 125.0, 126.4, 126.49, 126.55, 127.2, 127.3, 129.6, 130.1, 130.9, 131.3, 132.1, 133.4.

**X-ray crystallography**. Single crystals of compound 1, 1c, and 3 were prepared by a slow vapor diffusion of methanol into a concentrated solution of 1 or 1c in chloroform and a slow evaporation of toluene solution of 3 at room temperature. X-ray crystallographic analyses for 1c and 3 was performed on a Rigaku Saturn724+ CCD diffractometer with a graphite-monochromated Mo Kα radiation (λ = 0.71075 Å). The data collection and cell refinement were performed using CrystalClear-SM Expert 2.1 b46 software (Rigaku, 2016). X-ray crystallographic analysis for 1 was performed on a Bruker Single Crystal CCD X-ray Diffractometer (SMART APEX II) with Mo Kα radiation (λ = 0.71073 Å). The data collection and cell refinement were performed using APEX3 software (v2016.9-0, Bruker AXS, 2016). The structure of 1 observed based on the diffraction data was refined as a two-component twin using Olex2 software. The all structures were solved by direct methods (SHELXT) and refined by a full-matrix least-squares techniques against $F^2$ (SHELXL). The all non-hydrogen atoms were refined anisotropically. Hydrogen atoms were placed using AFIX instructions.

## Data availability

All data supporting the findings of this study are included in this article and its Supplementary Information. The X-ray crystallographic data have been deposited at the Cambridge Crystallographic Data Centre under deposition numbers CCDC 2118093 (1), CCDC 2118109 (1c), and CCDC 2117834 (3). Copies of the data are available free of charge via https://www.ccdc.cam.ac.uk/structures/.

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

## Acknowledgements

This work was supported by a Grant-in-Aid for Transformative Research Areas (A) "Condensed Conjugation" (JSPS KAKENHI Grant Number JP20H05868) from MEXT, Japan (K.M.), and by JST, PRESTO Grant Number JPMJPR20AE, Japan (T.H.). Y.N. acknowledges JSPS for a Grant-in-Aid for JSPS Research Fellow (JSPS KAKENHI Grant Number JP19J21095). A part of the computation time was provided by the Super Computer System, Institute for Chemical Research, Kyoto University.

## Author contributions

T.H. and K.M. conceived and supervised the project. Y.N. and R.L. synthesized the target compounds and Y.N. performed photophysical and electrochemical measurements. Y.N., D.S., and T.H. solved the crystal structure and carried out theoretical calculations. H.S. and H.M. measured and analyzed the time-resolved transient absorption measurements. Y.N., M.G., and J.-y.H. calculated and analyzed the structures at the conical intersections. Y.N., T.H., and K.M. wrote the manuscript. All the authors discussed the results and contributed to the manuscript.

## Competing interests

The authors declare no competing interests.
