## [Peer Review File · Nature Communications]

REVIEWER COMMENTS

Reviewer #1 (Remarks to the Author):

This is an interesting manuscript which describes the synthesis of per-peri-perbenzo[5]- and [9]helicenes, as well as that pi-extended [7]helicene previously reported by the same group.

The most significant finding and novelty in this work is that the studied pi-extended helicenes are considered helically fused oligo-phenanthrenes. This is an interesting observation since the HOMO–LUMO gap in these systems decreased from 2.14 to 1.15 eV by increasing the helical length. This experimental finding has been accounted for by the large effective conjugation length (ECL) of the pi-extended helical framework. In particular, and interestingly, this large ECL stems from the orbital interactions existing between the phenanthrene subunits at the 9- and 10-positions, thus forming a polyene-like moiety.

DFT calculations nicely confirm the observed ultrafast decay dynamics on the sub-picosecond timescale, which has been accounted for by the calculated low-lying conical intersection.

Finally, regarding the photophysical characterization of the new molecules, it seems to have been performed carefully and the new results are coherent with the theoretical calculations and the previous literature.

In summary, I feel that this is an interesting piece of paper which could meet the criteria of novelty and quality to be accepted for publication in Nat. Commun. after addressing the following minor points:

1. Although the authors have previously reported a strongly related molecules, namely pi-extended [7]helicene (ref. 33), the results now reported have led to the interesting concept of the internal polyene with amazing electronic features which, certainly, is relevant in terms of conducting properties in curved systems, namely helicenes when compared with planar systems.
2. The authors claim that since phenanthrene has the largest orbital coefficients at the 9- and 10-positions (C=C double bond character), the strongest electronic coupling is obtained when phenanthrene are covalently connected together, thus forming the helical phenanthrene oligomers. Since this fact significantly reduce the scope of this finding, perhaps the authors should provide, may be at the conclusions section, the perspectives of the results now reported in terms of the creation of other helically twisted molecular wires.
3. Please, make the following corrections:
 - Page 7, 2nd paragraph, 3rd line: overannulation (not overannuration).
 - Several times throughout the Manuscript and the Supporting Information the term "Transition absorption spectrum/a" is used (e.g., Figure 5 (a) caption. Please, use the term "Transient absorption spectrum/a".

Reviewer #2 (Remarks to the Author):

In brief, this is a nice piece of creative chemistry and valuable addition to the currently hot topic of extended helicenes. It offers various quite independent pieces of novelty, from synthesis to bonding situation to optics, so that I have no hesitation to recommend publication. While conceptually, there is some precedence on the synthetic side due to the authors' work on the [7]helicene, the availability of the homologues series of [5],[7] and [9] is essential and has still defined major synthetic challenges. I am intrigued by the successful photocyclization

I can stay away from summarizing the key take-home messages. What I should like to do instead is to raise a few critical issues. These should prompt the authors to reconsider some points, but should not cast any doubt at my positive recommendation !

The doubly linked helical oligomers with phenanthrene units establish the inner polyene structure which is proven by firm evidence. The discussion of the ECL is instructive. In the rylene series, comments on the „polyperylene“ as end-point have played an important role, even more so, since they could later be made by on-surface synthesis as graphene nanoribbons. Some discussion on the extrapolation would add an extra kick.

The extremely short lifetimes of the S₁-states are related to the polyene character. That might need some more in-depth discussion, in particular, by comparison with other helicenes.

In larger and larger nanographenes with K-regions there was reliable proof of local aromaticities, but attempts at chemical evidence, e.g. by addition reactions, failed. Is anything known, even if it might be a problem of quantities, about the reactivity?

On page 4 there is this sentence „... molecules with large ECLs are potentially applicable to conductive materials in the nanometer scale, which are called molecular wires“. I do not fully understand this sentence, although it must be admitted that we may be partially responsible for this confusion. The concept applies to single molecule experiments which raises many additional questions of interfacing with electrodes, nature of the charge carriers, contribution of tunneling etc. My question is whether this concept is needed here, even if it is also placed on top of the following figure. Conductivity is never raised later, neither the interfacing, whereas the optics plays such a prominent role.

Some typos should be corrected, see for example in the last section on pg 7, „overannuration“.

I have enjoyed reading this manuscript.

Reply to Reviewer #1

Ms.No: NCOMMS-21-46674-T

Title: "Doubly linked chiral phenanthrene oligomers for homogeneously π -extended helicenes with large effective conjugation length"

Authors: Yusuke Nakakuki, Takashi Hirose, Hikaru Sotome, Min Gao, Daiki Shimizu, Ruiji Li, Jun-ya Hasegawa, Hiroshi Miyasaka, and Kenji Matsuda

Thank you for reviewing our manuscript carefully. According to your comments we have revised our manuscript as follows:

Comment (1) This is an interesting manuscript which describes the synthesis of *per-peri*-perbenzo[5]- and [9]helicenes, as well as that π -extended [7]helicene previously reported by the same group. The most significant finding and novelty in this work is that the studied π -extended helicenes are considered helically fused oligo-phenanthrenes. This is an interesting observation since the HOMO–LUMO gap in these systems decreased from 2.14 to 1.15 eV by increasing the helical length. This experimental finding has been accounted for by the large effective conjugation length (ECL) of the π -extended helical framework. In particular, and interestingly, this large ECL stems from the orbital interactions existing between the phenanthrene subunits at the 9- and 10-positions, thus forming a polyene-like moiety. DFT calculations nicely confirm the observed ultrafast decay dynamics on the sub-picosecond timescale, which has been accounted for by the calculated low-lying conical intersection. Finally, regarding the photophysical characterization of the new molecules, it seems to have been performed carefully and the new results are coherent with the theoretical calculations and the previous literature.

In summary, I feel that this is an interesting piece of paper which could meet the criteria of novelty and quality to be accepted for publication in *Nat. Commun.* after addressing the following minor points:

Although the authors have previously reported a strongly related molecules, namely π -extended [7]helicene (ref. 33), the results now reported have led to the interesting concept of the internal polyene with amazing electronic features which, certainly, is relevant in terms of conducting properties in curved systems, namely helicenes when compared with planar systems.

Answer (1) Thank you for reviewing our manuscript carefully and giving positive comments that our manuscript is of sufficient novelty and quality to be published in *Nat. Commun.*

Comment (2) The authors claim that since phenanthrene has the largest orbital coefficients at the 9- and 10-positions (C=C double bond character), the strongest electronic coupling is obtained when phenanthrene are covalently connected together, thus forming the helical phenanthrene oligomers. Since this fact significantly reduce the scope of this finding, perhaps the authors should provide, may be at the conclusions section, the perspectives of the results now reported in terms of the creation of other helically twisted molecular wires.

Answer (2) We thank the reviewer for this insightful comment. In accordance with this comment, we have added the following sentence in the conclusions section on page 20.

"The creation of other π -extended HPAHs with different spatial distributions of frontier orbitals, e.g., HPAHs having a large ECL with their frontier orbitals localized at the outer rim moieties, will expand the scope of the chemistry of helical molecular wires."

We are curious about the relationship between the spatial distribution of frontier molecular orbitals in π -extended HPAHs and their carrier transport properties as molecular wires, hopefully with an efficient spin selectivity due to the helical chirality. To clarify this point, further intensive studies are needed. We believe that the results in this work will lead to further development of the chemistry of helical molecular wires in the future.

Comment (3) Please, make the following corrections:

- Page 7, 2nd paragraph, 3rd line: overannulation (not overannuration).
- Several times throughout the Manuscript and the Supporting Information the term "Transition absorption spectrum/a" is used (e.g., Figure 5 (a) caption. Please, use the term "Transient absorption spectrum/a".

Answer (3) We thank the reviewer for the comment. The misspellings of the terms "overannulation" and "Transient absorption spectra" have been corrected throughout the manuscript.

Reply to Reviewer #2

Ms.No: NCOMMS-21-46674-T

Title: "Doubly linked chiral phenanthrene oligomers for homogeneously π -extended helicenes with large effective conjugation length"

Authors: Yusuke Nakakuki, Takashi Hirose, Hikaru Sotome, Min Gao, Daiki Shimizu, Ruiji Li, Jun-ya Hasegawa, Hiroshi Miyasaka, and Kenji Matsuda

Comment (1) In brief, this is a nice piece of creative chemistry and valuable addition to the currently hot topic of extended helicenes. It offers various quite independent pieces of novelty, from synthesis to bonding situation to optics, so that I have no hesitation to recommend publication. While conceptually, there is some precedence on the synthetic side due to the authors' work on the [7]helicene, the availability of the homologues series of [5],[7] and [9] is essential and has still defined major synthetic challenges. I am intrigued by the successful photocyclization

I can stay away from summarizing the key take-home messages. What I should like to do instead is to raise a few critical issues. These should prompt the authors to reconsider some points, but should not cast any doubt at my positive recommendation!

Answer (1) Thank you for reviewing our manuscript carefully and appreciating the importance of this work. We are grateful for your positive recommendation of this manuscript to be published in *Nat. Commun.*

Comment (2) The doubly linked helical oligomers with phenanthrene units establish the inner polyene structure which is proven by firm evidence. The discussion of the ECL is instructive. In the rylene series, comments on the „polyperylene“ as end-point have played an important role, even more so, since they could later be made by on-surface synthesis as graphene nanoribbons. Some discussion on the extrapolation would add an extra kick.

Answer (2) We thank the reviewer for this comment. As suggested by this comment, the electronic properties of the helical molecular wires with sufficient length is of great interest to discuss the limit of the elongation of the ECL and the properties of molecular wires with infinite length. In accordance with this comment, we have added the following sentence in the conclusions section on page 20.

"The development of helical molecular wires with sufficient length is of potential interest, as graphene nanoribbons with sufficient length have been experimentally achieved by on-surface synthesis."

We carefully discussed whether to quantify the slope and intercept values obtained by extrapolation of the plot of the excitation energy obtained by the DFT calculations against the reciprocal of the number of π -electrons shown in Figure 4b, but decided to refrain from reporting the quantitative values due to the lack of experimental evidence at this moment. It is noted that the π -extended [5]helicene (**1**) is unfortunately not suitable to determine the values of the slope and intercept because the molecular orbital distribution of **1** is significantly different from the other longer derivatives **2** and **3** (Figure 4). We will continue our research work towards the synthesis of the longer HPAHs derivatives, i.e., π -extended [11]- and [13]helicenes, to experimentally

demonstrate the relationship between molecular wire properties and the helical lengths.

Comment (3) The extremely short lifetimes of the S₁-states are related to the polyene character. That might need some more in-depth discussion, in particular, by comparison with other helicenes.

Answer (3) We thank the reviewer for this important comment. In accordance with this comment, we have added the following discussion on the comparison of the non-radiative decay rate constants (k_{nr}) with some recently reported π -extended helicenes. The reported π -extended helicenes without the polyene character have the k_{nr} values on the order of 10^7 to 10^8 s⁻¹, which are 4 orders of magnitude slower than those of the π -extended helicenes with the polyene character, i.e. the k_{nr} values were $(0.83\text{--}2.6) \times 10^{12}$ s⁻¹ for **1**, **2**, and **3**. We believe that this comparison further supports the extremely short lifetimes of the S₁ state likely caused by the polyene-like electronic character of the π -extended helicenes.

We have added the following sentences in the main text on page 18, line 5,

"It is noted that recently reported π -extended helicenes, which do not have the polyene-like electronic structure along the inner helical rim, exhibit emission properties with the k_{nr} values on the order of 10^7 to 10^8 s⁻¹ (see Table S8 in Supplementary Information for details)."

Accordingly, we have added Table S8 and references S15–S18 in the Supplementary Information as follows.

Table S8. Emission properties of recently reported π -extended helicenes

Compd.	Solvent	$\lambda_{em,max}$ / nm	Φ_f	$\langle \tau_{S1} \rangle$ / ns	k_f / ns ⁻¹	k_{nr} / ns ⁻¹
A ^{S15}	CH ₂ Cl ₂	610	0.098	18	0.005	0.050
B ^{S16}	CH ₂ Cl ₂	870	0.046	14	0.003	0.068
C ^{S17}	CH ₂ Cl ₂	697	0.035	1	0.04	0.9
D ^{S18}	THF	528	0.41	7.1	0.058	0.083

- (S15) Cruz, C. M., Castro-Fernández, S., Maçôas, E., Cuerva, J. M. & Campaña, A. G. Undecabenzof[7]superhelicene: A Helical Nanographene Ribbon as a Circularly Polarized Luminescence Emitter. *Angew. Chem. Int Ed.* **57**, 14782–14786 (2018).
- (S16) Wang, Y., Yin, Z., Zhu, Y., Gu, J., Li, Y. & Wang, J. Hexapole [9]Helicene. *Angew. Chem. Int Ed.* **58**, 587–591 (2019).
- (S17) Hu, Y., Paternò, G. M., Wang, X.-Y., Wang, X.-C., Guizzardi, M., Chen, Q., Schollmeyer, D., Cao, X.-Y., Cerullo, G., Scotognella, F., Müllen, K. & Narita, A. π -Extended Pyrene-Fused Double [7]Carbohelicene as a Chiral Polycyclic Aromatic Hydrocarbon. *J. Am. Chem. Soc.* **141**, 12797–12803 (2019).
- (S18) Qiu, Z., Ju, C.-W., Frédéric, L., Hu, Y., Schollmeyer, D., Pieters, G., Müllen, K. & Narita, A. Amplification of Dissymmetry Factors in p-Extended [7]- and [9]Helicenes. *J. Am. Chem. Soc.* **143**, 4661–4667 (2021).

Comment (4) In larger and larger nanographenes with K-regions there was reliable proof of local aromaticities, but attempts at chemical evidence, e.g. by addition reactions, failed. Is anything known, even if it might be a problem of quantities, about the reactivity?

Answer (4) We thank the reviewer for this comment. The investigation on the chemical reactivity and post-functionalization of the π -extended helicenes is an ongoing project in our group, which is interesting but challenging due to the difficulties in synthesizing large quantities of substrates via multiple reaction steps.

As a preliminary result, we detected a Diels-Alder reaction of the π -extended [7]helicene (**2**) with a benzyne derivative as a dienophile proceeded at the outer rim moiety of **2**. Hopefully, we will report the results on the post-functionalization of the π -extended helicenes in the near future. Unfortunately, due to the lack of sufficient experimental evidence for the exact chemical structure of the products at this moment, we have not revised the main text or Supplementary Information in response to this comment.

Comment (5) On page 4 there is this sentence „... molecules with large ECLs are potentially applicable to conductive materials in the nanometer scale, which are called molecular wires“. I donnot fully understand this sentence, although it must be admitted that we may be partially responsible for this confusion. The concept applies to single molecule experiments which raises many additional questions of interfacing with electrodes, nature of the charge carriers, contribution of tunneling etc. My question is whether this concept is needed here, even if it is also placed on top of the following figure. Conductivity is never raised later, neither the interfacing, whereas the optics plays such a prominent role.

Answer (5) We thank the reviewer for this very important comment. This confusion likely stems from the fact that there seems to be no clear direct correlation between the large ECLs and the carrier conductivity. What we intend to discuss in this work is the expected conductive properties inside the π -conjugated molecules, but not the conductivity between the electrodes of single-molecule junctions connecting a conductive molecule. In order to avoid the confusion, we have added the following sentence in the main text on page 4, at the beginning of the second paragraph. It now reads as follows.

"Considering the transport of charge carriers along a covalently bonded organic π -

conjugated system, large interactions between the π -electrons of adjacent units are the key to good carrier conductivity. In this context, π -conjugated molecules with large ECLs are potentially applicable to conductive materials in the nanometer scale, which are called molecular wires.^{21,22}

We agree with the reviewer that the carrier conductivity between the electrodes of the single-molecule junctions depends on the nature of interface with the electrodes, the types of carriers, and contribution of resonant tunneling, as reported in literatures. Among many parameters, we believe that the molecular conductivity inside the π -conjugated molecules is necessary for an efficient performance of single-molecule junctions with long interelectrode gaps of several nanometers. The present results demonstrate the logical molecular design of π -extended HPAHs with a significantly small HOMO–LUMO gap. We believe that this work will lead to further development of the chemistry of helical molecular wires in the future.

Comment (6) Some typos should be corrected, see for example in the last section on pg 7, „overannuration“.

Answer (6) We thank the reviewer for the comment. The misspellings of the term “overannulation” have been corrected throughout the manuscript.

REVIEWERS' COMMENTS

Reviewer #1 (Remarks to the Author):

The authors have nicely addressed my previous concerns and, therefore, the manuscript meets the criteria of novelty and quality to be accepted for publication in Nature Communications as it stands.

Reviewer #2 (Remarks to the Author):

The authors have obviously taken great care in addressing the reviewer's comments and the paper can now be published at it stands. The authors had really done a good job and improved the manuscript.

Reply to Reviewer #1

Ms.No: NCOMMS-21-46674A

Title: "Doubly linked chiral phenanthrene oligomers for homogeneously π -extended helicenes with large effective conjugation length"

Authors: Yusuke Nakakuki, Takashi Hirose, Hikaru Sotome, Min Gao, Daiki Shimizu, Ruiji Li, Jun-ya Hasegawa, Hiroshi Miyasaka, and Kenji Matsuda

Comment (1) The authors have nicely addressed my previous concerns and, therefore, the manuscript meets the criteria of novelty and quality to be accepted for publication in Nature Communications as it stands.

Answer (1) We thank the reviewer for the remark and his/her recommendation for publication.

Reply to Reviewer #2

Ms.No: NCOMMS-21-46674A

Title: "Doubly linked chiral phenanthrene oligomers for homogeneously π -extended helicenes with large effective conjugation length"

Authors: Yusuke Nakakuki, Takashi Hirose, Hikaru Sotome, Min Gao, Daiki Shimizu, Ruiji Li, Jun-ya Hasegawa, Hiroshi Miyasaka, and Kenji Matsuda

Comment (1) The authors have obviously taken great care in addressing the reviewer's comments and the paper can now be published at it stands. The authors had really done a good job and improved the manuscript.

Answer (1) Thank you for reviewing our manuscript carefully and appreciating the importance of this work.